# A Probabilistic Framework for Multimodal Retrieval using Integrative Indian Buffet Process

**Bahadir Ozdemir**
Department of Computer Science
University of Maryland
College Park, MD 20742 USA
ozdemir@cs.umd.edu

**Larry S. Davis**
Institute for Advanced Computer Studies
University of Maryland
College Park, MD 20742 USA
lsd@umiacs.umd.edu

## Abstract

We propose a multimodal retrieval procedure based on latent feature models. The procedure consists of a Bayesian nonparametric framework for learning underlying semantically meaningful abstract features in a multimodal dataset, a probabilistic retrieval model that allows cross-modal queries and an extension model for relevance feedback. Experiments on two multimodal datasets, PASCAL-Sentence and SUN-Attribute, demonstrate the effectiveness of the proposed retrieval procedure in comparison to the state-of-the-art algorithms for learning binary codes.

## 1 Introduction

As the number of digital images which are available online is constantly increasing due to rapid advances in digital camera technology, image processing tools and photo sharing platforms, similarity-preserving binary codes have received significant attention for image search and retrieval in large-scale image collections [1, 2]. Encoding high-dimensional descriptors into compact binary strings has become a very popular representation for images because of their high efficiency in query processing and storage capacity [3, 4, 5, 6].

The most widely adapted strategy for similarity-preserving binary codes is to find a projection of data points from the original feature space to Hamming space. A broad range of hashing techniques can be categorized as data independent and dependent schemes. Locality sensitive hashing [3] is one of the most widely known data-independent hashing techniques. This technique has been extended to various hashing functions with kernels [4, 5]. Notable data-dependent hashing techniques include spectral hashing [1], iterative quantization [6] and spherical hashing [7]. Despite the increasing amount of multimodal data, especially in multimedia domains e.g. images with tags, most existing hashing techniques, unfortunately, focus on unimodal data. Hence, they inevitably suffer from the semantic gap, which is defined in [8] as the lack of coincidence between low level visual features and high level semantic interpretation of an image. On the other hand, joint analysis of multimodal data offers improved search and cross-view retrieval capabilities e.g. text-to-image queries by bridging the semantic gap. However, it also poses challenges associated with handling cross-view similarity.

Most recent studies have concentrated on multimodal hashing. Bronstein *et al*. proposed cross-modality similarity learning via a boosting procedure [9]. Kumar and Udupa presented a cross-view similarity search [10] by generalizing spectral hashing [1] for multi-view data objects. Zhen and Yeung described two recent methods: Co-regularized hashing [11] based on a boosted co-regularization framework and a probabilistic generative approach called multimodal latent binary embedding [12] based on binary latent factors. Nitish and Salakhutdinov proposed a deep Boltzmann machine for multimodal data [13]. Recently, Rastegari *et al*. proposed a predictable dual-view hashing [14] that aims to minimize the Hamming distance between binary codes obtained from two different views by utilizing multiple SVMs. Most of the multimodal hashing techniques are computationally ex-

pensive, especially when dealing with large-scale data. High computational and storage complexity restricts their scalability.

Although many hashing approaches rely on supervised information like semantic class labels, class memberships are not available for many image datasets. In addition, some supervised approaches cannot be generalized to unseen classes that are not used during training [15] even though new classes emerge in the process of adding new images to online image databases. Besides, every user's need is different and time varying [16]. Therefore, user judgments indicating the relevance of an image retrieved for a query are utilized to achieve better retrieval performance in the revised ranking of images [17]. Development of an efficient retrieval system that embeds information from multiple domains into short binary codes and takes relevance feedback into account is quite challenging.

In this paper, we propose a multimodal retrieval method based on latent features. A probabilistic approach is employed for learning binary codes, and also for modeling relevance and user preferences in image retrieval. Our model is built on the assumption that each image can be explained by a set of semantically meaningful *abstract features* which have both visual and textual components. For example, if an image in the dataset contains a side view of a car, the words "car", "automobile" or "vehicle" will probably appear in the description; also an object detector trained for vehicles will detect the car in the image. Therefore, each image can be represented as a binary vector, with entries indicating the presence or absence of each abstract feature.

Our contributions can be summarized in three aspects:

1. We propose a Bayesian nonparametric framework based on the Indian Buffet Process (IBP) [18] for integrating multimodal data in a latent space. Since the IBP is a nonparametric prior in an infinite latent feature model, the proposed method offers a flexible way to determine the number of underlying abstract features in a dataset.

2. We develop a retrieval system that can respond to cross-modal queries by introducing new random variables indicating relevance to a query. We present a Markov chain Monte Carlo (MCMC) algorithm for inference of the relevance from data.

3. We formulate relevance feedback as pseudo-images to alter the distribution of images in the latent space so that the ranking of images for a query is influenced by user preferences.

The rest of the paper is organized as follows: Section 2 describes the proposed integrative procedure for learning binary codes, retrieval model and processing relevance feedback in detail. Performance evaluation and comparison to state-of-the-art methods are presented in Section 3, and Section 4 provides conclusions.

## 2   Our Approach

In our data model, each image has both textual and visual components. To facilitate the discussion, we assume that the dataset is composed of two full matrices; our approach can easily handle images with only one component and it can be generalized to more than two modalities as well. We denote the data in the textual and visual space by $\mathbf{X}^\tau$ and $\mathbf{X}^v$, respectively. $\mathbf{X}^*$ is an $N \times D^*$ matrix whose rows corresponds to images in either space where $*$ is a placeholder used for either $v$ or $\tau$. The values in each column of $\mathbf{X}^*$ are centered by subtracting the sample mean of that column. The dimensionality of the textual space $D^\tau$ and the dimensionality of the visual space $D^v$ can be different. We use $\mathcal{X}$ to represent the set $\{\mathbf{X}^\tau, \mathbf{X}^v\}$.

### 2.1   Integrative Latent Feature Model

We focus on how textual and visual values of an image are generated by a linear-Gaussian model and its extension for retrieval systems. Given a multimodal image dataset, the textual and visual data matrices, $\mathbf{X}^\tau$ and $\mathbf{X}^v$, can be approximated by $\mathbf{Z}\mathbf{A}^\tau$ and $\mathbf{Z}\mathbf{A}^v$, respectively. $\mathbf{Z}$ is an $N \times K$ binary matrix where $Z_{nk}$ equals to one if abstract feature $k$ is present in image $n$ and zero otherwise. $\mathbf{A}^*$ is a $K \times D^*$ matrix where the textual and visual values for abstract feature $k$ are stored in row $k$ of $\mathbf{A}^\tau$ and $\mathbf{A}^v$, respectively (See Figure 1 for an illustration). The set $\{\mathbf{A}^\tau, \mathbf{A}^v\}$ is denoted by $\mathcal{A}$.

Our initial goal is to learn abstract features present in the dataset. Given $\mathcal{X}$, we wish to compute the posterior distribution of $\mathbf{Z}$ and $\mathcal{A}$ using Bayes' rule

$$p(\mathbf{Z}, \mathcal{A}|\mathcal{X}) \propto p(\mathbf{X}^\tau|\mathbf{Z}, \mathbf{A}^\tau)p(\mathbf{A}^\tau)p(\mathbf{X}^v|\mathbf{Z}, \mathbf{A}^v)p(\mathbf{A}^v)p(\mathbf{Z}) \tag{1}$$

where $\mathbf{Z}$, $\mathbf{A}^\tau$ and $\mathbf{A}^v$ are assumed to be a priori independent. In our model, the vectors for textual and visual properties of an image are generated from Gaussian distributions with covariance matrix $(\sigma_x^*)^2 \mathbf{I}$ and expectation $\mathbb{E}[\mathbf{X}^*]$ equal to $\mathbf{Z}\mathbf{A}^*$. Similarly, a prior on $\mathbf{A}^*$ is defined to be Gaussian with zero mean vector and covariance matrix $(\sigma_a^*)^2 \mathbf{I}$. Since we do not know the exact number of abstract features present in the dataset, we employ the Indian Buffet Process (IBP) to generate $\mathbf{Z}$, which provides a flexible prior that allows $K$ to be determined at inference time (See [18] for details). The graphical model of our integrative approach is shown in Figure 2.

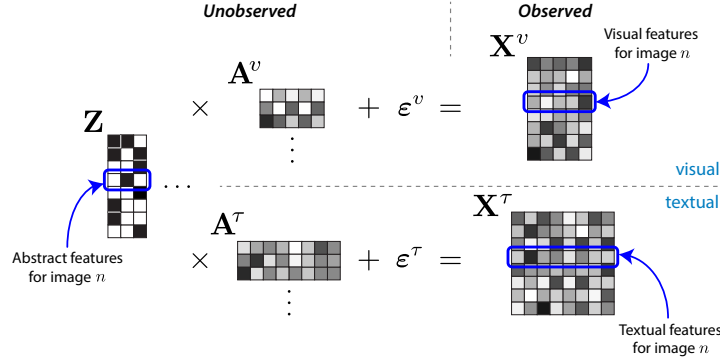

Figure 1: The latent abstract feature model proposes that visual data $\mathbf{X}^v$ is a product of $\mathbf{Z}$ and $\mathbf{A}^v$ with some noise; and similarly the textual data $\mathbf{X}^\tau$ is a product of $\mathbf{Z}$ and $\mathbf{A}^\tau$ with some noise.

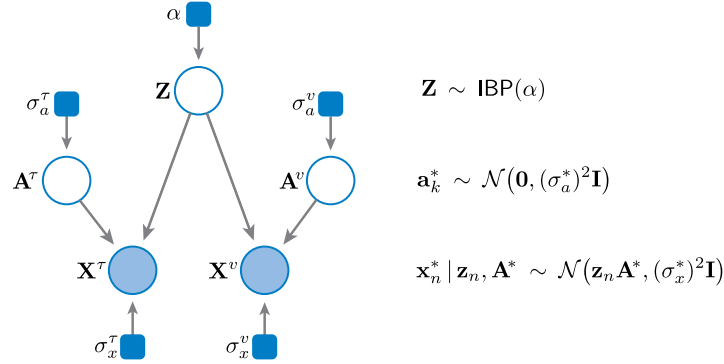

Figure 2: Graphical model for the integrative IBP approach where circles indicate random variables, shaded circles denote observed values, and the blue square boxes are hyperparameters.

The exchangeability property of the IBP leads directly to a Gibbs sampler which takes image $n$ as the last customer to have entered the buffet. Then, we can sample $Z_{nk}$ for all initialized features $k$ via

$$p(Z_{nk} = 1|\mathbf{Z}_{-nk}, \mathcal{X}) \propto p(Z_{nk} = 1|\mathbf{Z}_{-nk})p(\mathcal{X}|\mathbf{Z}). \tag{2}$$

where $\mathbf{Z}_{-nk}$ denotes entries of $\mathbf{Z}$ other than $Z_{nk}$. In the finite latent feature model (where $K$ is fixed), the conditional distribution for any $Z_{nk}$ is given by

$$p(Z_{nk} = 1|\mathbf{Z}_{-nk}) = \frac{m_{-n,k} + \frac{\alpha}{K}}{N + \frac{\alpha}{K}} \tag{3}$$

where $m_{-n,k}$ is the number of images possessing abstract feature $k$ apart from image $n$. In the infinite case like the IBP, we obtain $p(Z_{nk} = 1|\mathbf{Z}_{-nk}) = \frac{m_{-n,k}}{N}$ for any $k$ such that $m_{-n,k} > 0$. We also need to draw new features associated with image $n$ from $\mathrm{Poisson}\left(\frac{\alpha}{N}\right)$, and the likelihood term is now conditioned on $\mathbf{Z}$ with new additional columns set to one for image $n$.

For the linear-Gaussian model, the collapsed likelihood function $p(\mathcal{X}|\mathbf{Z}) = p(\mathbf{X}^\tau|\mathbf{Z})p(\mathbf{X}^v|\mathbf{Z})$ can be computed using

$$p(\mathbf{X}^*|\mathbf{Z}) = \int p(\mathbf{X}^*|\mathbf{Z}, \mathbf{A}^*)p(\mathbf{A}^*)\, d\mathbf{A}^* = \frac{\exp\left\{-\frac{1}{2(\sigma_x^*)^2}\operatorname{tr}\left(\mathbf{X}^{*T}(\mathbf{I} - \mathbf{Z}\mathbf{M}\mathbf{Z}^T)\mathbf{X}^*\right)\right\}}{(2\pi)^{\frac{ND^*}{2}}(\sigma_x^*)^{(N-K)D^*}(\sigma_a^*)^{KD^*}|\mathbf{M}|^{\frac{-D^*}{2}}} \tag{4}$$

where $\mathbf{M} = \left(\mathbf{Z}^T\mathbf{Z} + \frac{(\sigma_x^*)^2}{(\sigma_a^*)^2}\mathbf{I}\right)^{-1}$ and $\operatorname{tr}(\cdot)$ is the trace of a matrix [18]. To reduce the computational complexity, Doshi-Velez and Ghahramani proposed an accelerated sampling in [19] by maintaining the posterior distribution of $\mathbf{A}^*$ conditioned on partial $\mathbf{X}^*$ and $\mathbf{Z}$. We use this approach to learn binary codes, *i.e.* the feature-assignment matrix $\mathbf{Z}$, for multimodal data. Unlike the hashing methods that learn optimal hyperplanes from training data [6, 7, 14], we only sample $\mathbf{Z}$ without specifying the length of binary codes in this process. Therefore, the binary codes can be updated efficiently if new images are added in a long run of the retrieval system.

## 2.2 Retrieval Model

We extend the integrative IBP model for image retrieval. Given a query, we need to sort the images in the dataset with respect to their relevance to the query. A query can be comprised of textual and visual data, or either component can be absent. Let $\mathbf{q}^\tau$ be a $D^\tau$-dimensional vector for the textual values and $\mathbf{q}^v$ be a $D^v$-dimensional vector for the visual values of the query. We can write $\mathcal{Q} = \{\mathbf{q}^\tau, \mathbf{q}^v\}$. As for the images in $\mathcal{X}$, we consider a query to be generated by the same model described in the previous section with the exception of the prior on abstract features. In the retrieval part, we consider $\mathbf{Z}$ as a known quantity and we fix the number abstract features to $K$. Therefore, the feature-assignments for the dataset are not affected by queries. In addition, queries are explained by known abstract features only.

We extend the Indian restaurant metaphor to construct the retrieval model. A query corresponds to the $(N + 1)$th customer to enter the buffet. The previous customers are divided into two classes as friends and non-friends based on their relevance to the new customer. The new customer now samples from at most $K$ dishes in proportion to their popularity among friends and also their un-popularity among non-friends. Consequently, the dishes sampled by the new customer are expected to be similar to those of friends and dissimilar to those of non-friends. Let $\mathbf{r}$ be an $N$-dimensional vector where $r_n$ equals to one if customer $n$ is a friend of the new customer and zero otherwise. For this finitely long buffet, the sampling probability of dish $k$ by the new customer can be written as $\frac{m_k' + \alpha/K}{N + 1 + \alpha/K}$ where $m_k' = \sum_{n=1}^{N}(Z_{nk})^{r_n}(1 - Z_{nk})^{1-r_n}$, that is the total number of friends who tried dish $k$ and non-friends who did not sample dish $k$. Let $\mathbf{z}'$ be a $K$-dimensional vector where $z_k'$ records if the new customer (query) sampled dish $k$. We place a prior over $r_n$ as Bernoulli$(\theta)$. Then, we can sample $z_k'$ from

$$p(z_k' = 1|\mathbf{z}_{-k}', \mathcal{Q}, \mathbf{Z}, \mathcal{X}) \propto p(z_k' = 1|\mathbf{Z})p(\mathcal{Q}|\mathbf{z}', \mathbf{Z}, \mathcal{X}). \tag{5}$$

The probability $p(z_k' = 1|\mathbf{Z})$ can be computed efficiently for $k = 1, \ldots, K$ by marginalizing over $\mathbf{r}$ as below:

$$p(z_k' = 1|\mathbf{Z}) = \sum_{\mathbf{r} \in \{0,1\}^N} p(z_k' = 1|\mathbf{r}, \mathbf{Z})p(\mathbf{r}) = \frac{\theta m_k + (1 - \theta)(N - m_k) + \frac{\alpha}{K}}{N + 1 + \frac{\alpha}{K}}. \tag{6}$$

The collapsed likelihood of the query, $p(\mathcal{Q}|\mathbf{z}', \mathbf{Z}, \mathcal{X})$, is given by the product of textual and visual likelihood values, $p(\mathbf{q}^\tau|\mathbf{z}', \mathbf{Z}, \mathbf{X}^\tau)p(\mathbf{q}^v|\mathbf{z}', \mathbf{Z}, \mathbf{X}^v)$. If either textual or visual component is missing, we can simply integrate out the missing one by omitting the corresponding term from the equation. The likelihood of each part can be calculated as follows:

$$p(\mathbf{q}^*|\mathbf{z}', \mathbf{Z}, \mathbf{X}^*) = \int p(\mathbf{q}^*|\mathbf{z}', \mathbf{A}^*)p(\mathbf{A}^*|\mathbf{Z}, \mathbf{X}^*)\, d\mathbf{A}^* = \mathcal{N}(\mathbf{q}^*; \boldsymbol{\mu}_q^*, \boldsymbol{\Sigma}_q^*). \tag{7}$$

where the mean and covariance matrix of the normal distribution are given by $\boldsymbol{\mu}_q^* = \mathbf{z}'\mathbf{M}\mathbf{Z}^T\mathbf{X}^*$ and $\boldsymbol{\Sigma}_q^* = (\sigma_x^*)^2(\mathbf{z}'\mathbf{M}\mathbf{z}'^T + \mathbf{I})$, akin to the update equation in [19] (Refer to (4) for $\mathbf{M}$).

Finally, we use the conditional expectation of $\mathbf{r}$ to rank images in the dataset with respect to their relevance to the given query. Calculating the expectation $\mathbb{E}[\mathbf{r}|\mathcal{Q}, \mathbf{Z}, \mathcal{X}]$ is computationally expensive;

however, it can be empirically estimated using the Monte Carlo method as follows:

$$\hat{\mathbb{E}}[r_n|\mathcal{Q}, \mathbf{Z}, \mathcal{X}] = \frac{1}{I} \sum_{i=1}^{I} p(r_n = 1|\mathbf{z}'^{(i)}, \mathbf{Z}) = \frac{\theta}{I} \sum_{i=1}^{I} \prod_{k=1}^{K} \frac{p(z_k'^{(i)}|r_n = 1, \mathbf{Z})}{p(z_k'^{(i)}|\mathbf{Z})} \qquad (8)$$

where $\mathbf{z}'^{(i)}$ represents i.i.d. samples from (5) for $i = 1, \ldots, I$. The last equation required for computing (8) is

$$p(z_k' = 1|r_n = 1, \mathbf{Z}) = \frac{Z_{nk} + \theta m_{-n,k} + (1 - \theta)(N - 1 - m_{-n,k}) + \frac{\alpha}{K}}{N + 1 + \frac{\alpha}{K}}. \qquad (9)$$

The retrieval system returns a set of top ranked images to the user. Note that we compute the expectation of relevance vector instead of sampling directly since binary values indicating the relevance are less stable and they hinder the ranking of images.

## 2.3 Relevance Feedback Model

In our data model, user preferences can be described over abstract features. For instance, if abstract feature $k$ is present in the most of positive samples *i.e.* images judged as relevant by the user and it is absent in the irrelevant ones, then we can say that the user is more interested in the semantic subspace represented by abstract feature $k$. In the revised query, the images having abstract feature $k$ are expected to be ranked in higher positions in comparison to the initial query. We can achieve this desirable property from query-specific alterations to the sampling probability in (5) for the corresponding abstract features. Our approach is to add pseudo-images to the feature-assignment matrix $\mathbf{Z}$ before the computations of the revised query. For the Indian restaurant analogy, pseudo-images correspond to some additional friends of the new customer (query), who do not really exist in the restaurant. The distribution of dishes sampled by those imaginary customers reflects user relevance feedback. Thus, the updated expectation of the relevance vector has a bias towards user preferences.

Let $\mathbf{Z}_u$ be an $N_u \times K$ feature-assignment matrix for pseudo-images only; then the number of pseudo-images, $N_u$, determines the influence of relevance feedback. Therefore, we set an upper limit on $N_u$ as the number of real images, $N$, by placing a prior distribution as $N_u \sim \text{Binomial}(\gamma, N)$ where $\gamma$ is a parameter that controls the weight of feedback. Let $m_{u,k}$ be the number of pseudo-images containing abstract feature $k$; then this number has an upper bound $N_u$ by definition. For abstract feature $k$, a prior distribution conditioned on $N_u$ can be defined as $m_{u,k}|N_u \sim \text{Binomial}(\phi_k, N_u)$ where $\phi_k$ is a parameter that can be tuned by relevance judgments.

Let $\mathbf{z}''$ be a $K$-dimensional feature-assignment vector for the revised query; then we can sample each $z_k''$ via

$$p(z_k'' = 1|\mathbf{z}_{-k}'', \mathcal{Q}, \mathbf{Z}, \mathcal{X}) \propto p(z_k'' = 1|\mathbf{Z})p(\mathcal{Q}|\mathbf{z}'', \mathbf{Z}, \mathcal{X}) \qquad (10)$$

where the computation of the collapsed likelihood is already shown in (7). Note that we do not actually generate all entries of $\mathbf{Z}_u$ but only the sum of its columns $\mathbf{m}_u$ and number of rows $N_u$ for computing the sampling probability. We can write the first term as:

$$p(z_k'' = 1|\mathbf{Z}) = \sum_{N_u=0}^{N} p(N_u) \sum_{m_{u,k}=0}^{N_u} p(m_{u,k}|N_u) \sum_{\mathbf{r} \in \{0,1\}^N} p(z_k'' = 1|\mathbf{r}, \mathbf{Z}_u, \mathbf{Z})p(\mathbf{r})$$

$$= \sum_{j=0}^{N} \binom{N}{j} \gamma^j (1 - \gamma)^{N-j} \frac{\theta m_k + (1 - \theta)(N - m_k) + \frac{\alpha}{K} + \phi_k j}{N + 1 + \frac{\alpha}{K} + j} \qquad (11)$$

Unfortunately, this expression has no compact analytic form; however, it can be efficiently computed numerically by contemporary scientific computing software even for large values of $N$. In this equation, one can alternatively fix $r_n$ to 1 if the user marks observation $n$ as relevant or 0 if it is indicated to be irrelevant. Finally, the expectation of $\mathbf{r}$ is updated using (8) with new i.i.d. samples $\mathbf{z}''^{(i)}$ from (10) and the system constructs the revised set of images.

# 3 Experiments

The experiments were performed in two phases. We first compared the performance of our method in category retrieval with several state-of-the-art hashing techniques. Next, we evaluated the improvement in the performance of our method with relevance feedback. We used the same multimodal datasets as [14], namely PASCAL-Sentence 2008 dataset [20] and the SUN-Attribute dataset [21]. In the quantitative analysis, we used the mean of the interpolated precision at standard recall levels for comparing the retrieval performance. In the qualitative analysis, we present the images retrieved by our proposed method for a set of text-to-image and image-to-image queries. All experiments were performed in the Matlab environment[1].

## 3.1 Datasets

The PASCAL-Sentence 2008 dataset is formed from the PASCAL 2008 images by randomly selecting 50 images belonging to each of the 20 categories. In experiments, we used the precomputed visual and textual features provided by Farhadi *et al*. [20]. Amazon Mechanical Turk workers annotate five sentences for each of the 1000 images. Each image is labelled by a triplet of <*object, action, scene*> representing the semantics of the image from these sentences. For each image, the semantic similarity between each word in its triplet and all words in a dictionary constructed from the entire dataset is computed by the Lin similarity measure [22] using the WordNet hierarchy. The textual features of an image are the sum of all similarity vectors for the words in its triplet. Visual features are built from various object detectors, image classifiers and scene classifiers. These features contain the coordinates and confidence values that object detectors fire and the responses of image and scene classifiers trained on low-level image descriptors.

The SUN-Attribute dataset [21], a large-scale dataset of attribute-labeled scenes, is built on top of the existing SUN categorical dataset [23]. The dataset contains 102 attribute labels annotated by 3 Amazon Mechanical Turk workers for each of the 14,340 images from 717 categories. Each category has 20 annotated images. The precomputed visual features [21, 23] include gist, $2 \times 2$ histogram of oriented gradient, self-similarity measure, and geometric context color histograms. The attribute features is computed by averaging the binary labels from multiple annotators where each image is annotated with attributes from five types: materials, surface properties, functions or affordances, spatial envelope attributes and object presence.

## 3.2 Experimental Setup

Firstly, all features were centered to zero and normalized to unit length; also duplicate features were removed from the data. We reduced the dimensionality of visual features in the SUN dataset from 19,080 to 1,000 by random feature selection, which is preferable to PCA for preserving the variance among visual features. The Gibbs sampler was initialized with a randomly sampled feature assignment matrix $\mathbf{Z}$ from a IBP prior. We set $\alpha = 1$ in all experiments to keep binary codes short. The other hyperparameters $\sigma_a^*$ and $\sigma_x^*$ were determined by adding Metropolis steps to the MCMC algorithm in order to prevent one modality from dominating the inference process.

In the retrieval part, the relevance probability $\theta$ was set to $0.5$ so that all abstract features have equal prior probability from (6). Feature assignments of a query were initialized with all zero bits. For relevance feedback analysis, we set $\gamma = 1$ (equal significance for the data and feedback) and we decide each $\phi_k$ as follows:

Let $\bar{z}'_k = \frac{1}{I} \sum_{i=1}^{I} z_k'^{(i)}$ where each $\mathbf{z}'^{(i)}$ is drawn from (5) for a given query; and $\hat{z}'_k = \frac{1}{T} \sum_{t=1}^{T} (Z_{tk})^{r_t} (1 - Z_{tk})^{1-r_t}$ where $t$ represents the index of each image judged by the user and $T$ is the size of relevance feedback. The difference between these two quantities, $\delta_k = \bar{z}'_k - \hat{z}'_k$, controls $\phi_k$ which is defined by a logistic function as

$$\phi_k = \frac{1}{1 + e^{-(c\delta_k + \beta_{0,k})}} \tag{12}$$

where $c$ is a constant and $\beta_{0,k} = \ln \frac{p(z'_k=1|\mathbf{Z})}{p(z'_k=0|\mathbf{Z})}$ (refer to (6) for $p(z'_k|\mathbf{Z})$). We set $c = 5$ in our experiments. Note that $\phi_k = p(z'_k = 1|\mathbf{Z})$ when $\bar{z}'_k$ is equal to $\hat{z}'_k$.

### 3.3 Experimental Results

We compared our method, called integrative IBP (iIBP), with several hashing methods including locality sensitive hashing (LSH) [3], spectral hashing (SH) [1], spherical hashing (SpH) [7], iterative quantization (ITQ) [6], multimodal deep Boltzmann machine (mDBM) [13] and predictable dual-view hashing (PDH) [14]. We divided each dataset into two equal sized train and test segments. The train segment was first used for learning the feature assignment matrix $\mathbf{Z}$ by iIBP. Then, the other binary code methods were trained with the same code length $K$. We used supervised ITQ coupled with CCA [24] and took the dual-view approach [14] to construct basis vectors in a common subspace. However, LSH, SH and SpH were applied on single-view data since they do not support cross-view queries.

All images in the test segment were used as both image and text queries. Given a query, images in the train set were ranked by iIBP with respect to (8). For all other methods, we use Hamming distance between binary codes in the nearest-neighbor search. Mean precision curves are presented in Figure 3 for both datasets. Unlike the experiments in [14] performed in a supervised manner, the performance on the SUN-Attribute dataset is very low due to the small number of positive samples compared to the number of categories (Figure 3b). There are only 10 relevant images among 7,170 training images. Therefore, we also used Euclidean neighbor ground truth labels computed from visual data as in [6] (Figure 3c). As seen in the figure, our method (iIBP) outperforms all other methods. Although unimodal hashing methods perform well on text queries, they suffer badly on image queries because the semantic similarity to the query does not necessarily require visual similarity (Figures 3-4 in the supplementary material). By the joint analysis of visual and textual spaces, our approach improves the performance for image queries by bridging the semantic gap [8].

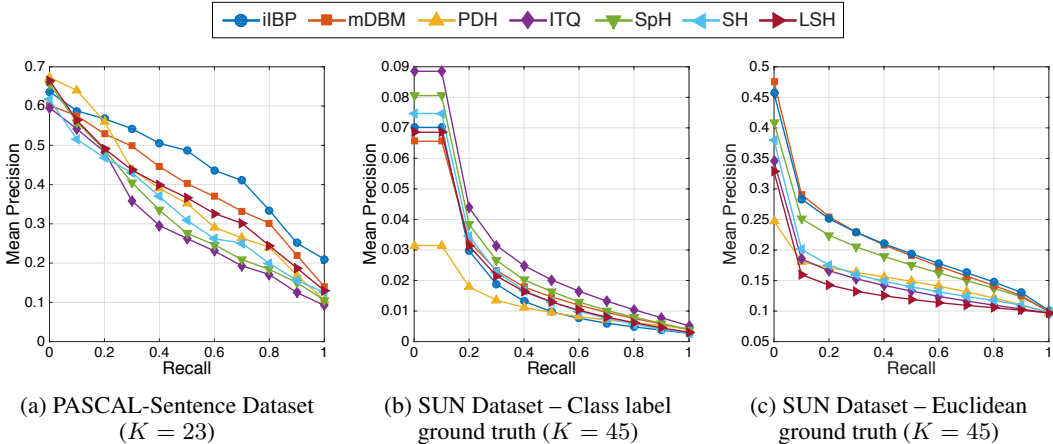

(a) PASCAL-Sentence Dataset ($K = 23$)  (b) SUN Dataset – Class label ground truth ($K = 45$)  (c) SUN Dataset – Euclidean ground truth ($K = 45$)

Figure 3: The result of category retrieval for all query types (image-to-image and text-to-image queries). Our method (iIBP) is compared with the-state-of-the-art methods.

For qualitative analysis, Figure 4a shows the top-5 retrieved images from the PASCAL-Sentence 2008 dataset for image queries. Thanks to the integrative approach, the retrieved images share remarkable semantic similarity with the query images. Similarly, most of the retrieved images for the text-to-image queries in Figure 4b comprise the semantic structure in the query sentences.

In the second phase of analyses, we utilized the rankings in the first phase to decide relevance feedback parameters independently for each query. We picked the top two relevant images as positive samples and top two irrelevant images as negative samples. We set each $\phi_k$ by (12) and reordered the images using the relevance feedback model excluding the ones used as user relevance judgements. Those images were omitted from precision-recall calculations as well. Figure 5 illustrates that relevance feedback slightly boosts the retrieval performance, especially for the PASCAL-Sentence dataset.

The computational complexity of an iteration is $O(K^2 + KD^*)$ for a query and $O(N(K^2 + KD^\tau + KD^v))$ for training [19]. The feature assignment vector $\mathbf{z}'$ of a query usually converges in a few

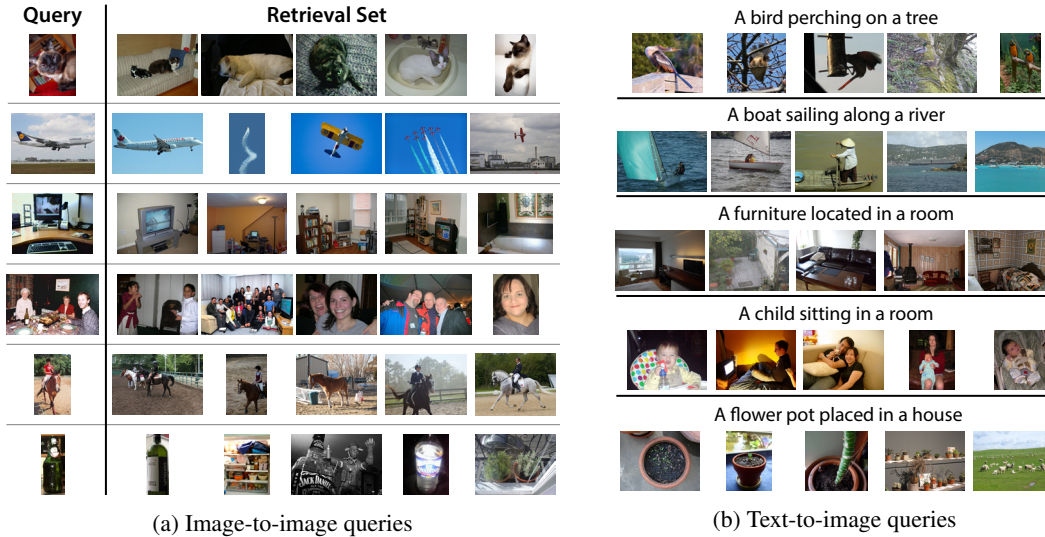

| Query | Retrieval Set |

(a) Image-to-image queries

A bird perching on a tree

A boat sailing along a river

A furniture located in a room

A child sitting in a room

A flower pot placed in a house

(b) Text-to-image queries

Figure 4: Sample images retrieved from the PASCAL-Sentence dataset by our method (iIBP)

iterations. A typical query took less than 1 second in our experiments for $I = 50$ with our optimized Matlab code.

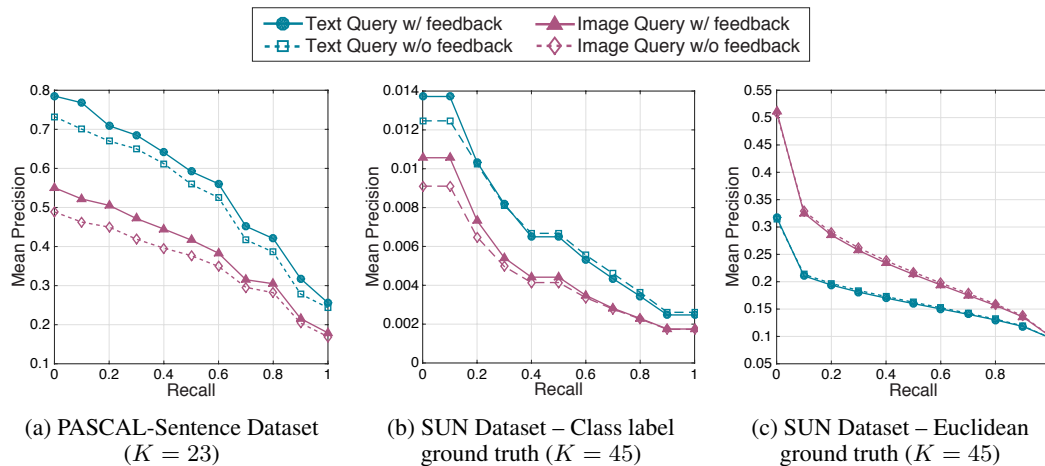

(a) PASCAL-Sentence Dataset
$(K = 23)$

(b) SUN Dataset – Class label
ground truth $(K = 45)$

(c) SUN Dataset – Euclidean
ground truth $(K = 45)$

Figure 5: The result of category retrieval by our approach (iIBP) with relevance feedback for text and image queries. Revised retrieval with relevance feedback is compared with initial retrieval.

# 4 Conclusion

We proposed a novel retrieval scheme based on binary latent features for multimodal data. We also describe how to utilize relevance feedback for better retrieval performance. The experimental results on real world data demonstrate that our method outperforms state-of-the-art hashing techniques. In our future work, we would like to develop a user inference to get relevance feedback and a deterministic variational method for inference the integrative IBP based on a truncated stick-breaking approximation.

### Acknowledgments

This work was supported by the NSF Grant 12621215 EAGER: Video Analytics in Large Heterogeneous Repositories.

## Footnotes

[1]Our code is available at http://www.cs.umd.edu/~ozdemir/iibp

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
