[Supplementary Material]

# Supplementary Material for
# A Probabilistic Framework for Multimodal Retrieval using Integrative Indian Buffet Process

**Bahadir Ozdemir**
Department of Computer Science
University of Maryland
College Park, MD 20742 USA
ozdemir@cs.umd.edu

**Larry S. Davis**
Institute for Advanced Computer Studies
University of Maryland
College Park, MD 20742 USA
lsd@umiacs.umd.edu

## 1 Our Approach

Figure 1 shows the schematic overview of our retrieval algorithm. Figure 2 demonstrates the graphical model of the relevance feedback model. Note that this model becomes equivalent to the query model when $\gamma = 0$.

Figure 1: Schematic overview of our retrieval algorithm. The flow chart illustrates discovery of abstract features from multimodal data, the retrieval system for cross-view queries and user relevance feedback.

Figure 2: Graphical model for the feedback query model. Circles indicate random variables, shaded circles denote observed values. Hyperparameters are omitted for clarity. Note that $\mathbf{Z}$ is considered as an observed variable in the retrieval part.

## 2 Experimental Results

Mean precision curves are presented in Figures 3 and 4 for text and image queries, respectively. The curves given in the main paper are the averages of these two types of queries. In addition, we analyzed the stability of our Gibbs sampler in retrieval from the PASCAL-Sentence dataset by running each query for 50 times. Figure 5 shows the range of mean precisions in trials by error bars for text and image queries. The error bars are very small at all recall levels for both query types. This concludes that the image set retrieved by our method for a given query is very stable. Figure 6 shows the effect of sample size in the Monte Carlo estimation, $I$ of $(8)$ in the main paper, on the retrieval performance for the PASCAL-Sentence dataset. The precision curves start to overlap at $I = 5$. Therefore, one can use a small sample set to estimate the expectation of relevance vector $\mathbf{r}$ for faster processing of the query at a similar precision level.

(a) PASCAL-Sentence Dataset ($K = 23$)

(b) SUN Dataset – Class label ground truth ($K = 45$)

(c) SUN Dataset – Euclidean ground truth ($K = 45$)

Figure 3: The result of category retrieval for text-to-image queries. Our method (iIBP) is compared with the-state-of-the-art methods.

(a) PASCAL-Sentence Dataset ($K = 23$)

(b) SUN Dataset – Class label ground truth ($K = 45$)

(c) SUN Dataset – Euclidean ground truth ($K = 45$)

Figure 4: The result of category retrieval for image-to-image queries. Our method (iIBP) is compared with the-state-of-the-art methods.

Figure 5: The result of stability analysis for text and image queries. Our method (iIBP) is applied on the PASCAL-Sentence dataset for 50 times. The curves represent the average level of mean precisions. Error bars indicate the range of mean precisions observed at each standard recall level.

Figure 6: The result of parameter effect analysis. Our method (iIBP) is applied on the PASCAL-Sentence dataset for different values of $I$, the sample size in the Monte Carlo estimation.