[Reviews · NeurIPS 2014]

Submitted by Assigned_Reviewer_21

This paper defines a joint generative model of an image and its annotated text which is used to learn a bit vector representation for large scale image retrieval. An Indian Buffet Process is used to learn the length of the bit vector. The method is compared favourably to several widely used techniques.

Quality
=======
It is good to see a retrieval paper constructed around a well-defined probabilistic model. The model is relatively simple - the assumption that X can be well approximated by ZA seems quite a strong one - it would be nice to see more discussion of the limitations imposed by this assumption. Given that Z is shared, unmodified, between the textual and visual halves of the model, it's not clear how well it would handle images with heavily occluded objects or text that doesn't mention some area of the scene. In addition, the use of a linear-Gaussian model means that the model is unimodal in visual space - which doesn't seem like a good idea. In general, there's quite a lot of discussion of the IBP which could be dropped in favour of more discussion of the model, since that is the main contribution of this paper.

The experiments are thorough and show that this approach adds value over other competing approaches. It would be nice to see more discussion about the reversal of ranking between Fig 3b and 3c.

Clarity
=======
The paper is well written and clearly presented. It would be helpful to include a brief summary of the Euclidean neighbor ground truth labels from [6], since I found it necessary to read that paper in order to understand the differences in relative performance of the proposed method in Fig 3b and Fig 3c.

Fig 2. would be made clearer by adding a plate over the images.

Typos
Line 038: "data-independent"->"data dependent"
Line 112: "priori"->"a priori"
Line 191: "As the images"->"As for the images"
Line 271: "scientic"->"scientific"

Originality & significance
==========================
The main contribution of this paper is the use of a new non-parametric Bayesian model for large scale image information retrieval that gives good performance compared to other commonly used methods. This is a much more rigorous treatment of the problem than previous more ad-hoc, algorithmic approaches. It also opens the doors to developing more sophisticated models in this space. The relevance feedback model also seems like a nice contribution.
Summary: This well-written paper describes a non-parametric Bayesian model of images and text for learning bit vectors for large scale image retrieval. It is rigorously defined and gives good results.

Submitted by Assigned_Reviewer_42

The paper proposes a probabilistic framework for multimodal image-text
retrieval. The model utilizes Indian Buffet Process (IBP) to infer
a set of latent binary features for every training example. These
features are then used to retrieve relevant images at query time.

The paper is very well written and the approach is interesting in
that it allows to dynamically grow the size of the latent
embeddings. I'm however concerned about the complexity of the
proposed method. If I correctly understood the retrieval model
in Section 2.2, it requires calculating expected relevance (eq. 8)
for every available training example. From this it seems that a full
pass through all the data is required for every query largely
defeating the purpose of Hamming mapping which allows for sublinear
nearest neighbor search.

Moreover, fairly involved Gibbs sampling (eq. 5) also needs to be
conducted for every query. This makes retrieval results
non-deterministic and potentially inconsistent where
different set of top matches can be returned when
the same query is re-issued. On line 402 you mention that only 25
samples were used to estimate the expectations, do you have any
variance/consistency results when this sampling procedure is
repeated multiple times for the same query?

Finally, there is very relevant recent work by Srivastava et al.,
NIPS2012, which explores the same multimodal image-text problem.
I think through empirical comparison with this work needs to be
conducted.
Summary: Overall I think that the paper proposes an interesting approach to the multimodal retrieval problem but lacks complexity justification and comparison with relevant existing work.

Submitted by Assigned_Reviewer_43

This paper proposes a framework for multimodal retrieval. A nonparametric Bayesian model is employed for representing latent semantic factors.

1. Model:
The proposed model may not be so novel, it is actually a collective matrix factorization model. The authors have augmented it as a Bayesian model by defining an Indian Buffet Process on the shared factorized matrix and a Gaussian distribution on each element of the other factorized matrices. With IBP, one can adaptively control the number of latent factors. Although the model may not be used for the task so far, it may not be difficult to come up with the idea.

The model appears to be reasonable. However, it is based on a strong assumption. Adding independent Gaussians on the elements of matrices and using linear transforms means to assume there exists independence between different dimensions of the observations.

2. Experiments:

Experiments are conducted on two too small datasets for a retrieval model, one dataset is 1000, and the other is 14340.

The authors should evaluate their model on a more challenging dataset, e.g. SBU Captioned Photo Dataset(1M). The baseline methods are not representative, the authors may want to compare to Ruslan Salakhutdinov's ICML2012 "Learning Representations for Multimodal Data with Deep Belief Nets"

Summary: The paper proposes a new framework for multimodal retrieval using Integrative Indian Buffet Process. The paper is well written and the work is solid.

However, I have the following concern. The idea may not be so novel. The experiments are not so strong. The proposed method is based on a strong assumption.
Author Feedback
Author rebuttal: We sincerely thank all reviewers for the very helpful comments and constructive suggestions on our manuscript. We address the common concerns first; then we list the individual responses to each reviewer below:

The model: We made our assumptions about the model with the aim of obtaining an image retrieval system that responds to queries efficiently. Despite its limitations, we prefer a simpler model to be learned from data using Gibbs sampling in this work. However, we plan to address the issues pointed out by the reviewers in our future work by taking a statistical approach to discover the structural forms between the abstract features; and also complex relationships between visual and textual modalities. We feel that these will strengthen the model significantly.

Comparison with Srivastava & Salakhutdinov: We thank the reviewers for referring this multimodal deep learning technique. We focused on binary hashing methods for comparison with our method. Thus, we did not include this recent publication. We could not finish conducting the empirical comparison during the rebuttal period; however, the revised version of the paper will include this among the baseline methods.

To Assigned_Reviewer_21:

The reversal of ranking between Fig 3: The textual description represents the semantics of an image (ground truth labels) better than visual counterpart; however, the Euclidean nearest neighbors were computed from visual space. Therefore, the performance for image queries is better in Fig 3c. We thank the reviewer for the suggestions for a clear presentation. We will add this explanation to the manuscript and make the other changes/corrections suggested by the reviewer in the revised version.

To Assigned_Reviewer_42:

Complexity of a query: Given a query, we first sample binary codes from (eq. 5) for I times with different initializations. The complexity of this part is independent of the size of dataset (N) but it depends on the number of latent features (K). Next, we use these I binary codes in (eq. 8) to compute an approximation to the expected relevance for every image in the database and this has a complexity of O(N*K) where K < < D. Therefore, it is faster than computing the Euclidean distance in very high dimensional datasets. On the other hand, setting the parameters theta=1 and I=1 in (eq. 8) makes our model equivalent to binary mapping using IBP. In this case, a sublinear nearest neighbor search can be employed. However, we observe that retrieval performance is improved using our parameter settings. (MAP score for the Pascal dataset increases from 0.42 to 0.49). We also repeated the experiments for 10 times to analyze the stability of results. MAP scores for the Pascal dataset vary from 0.490 to 0.494. We will add new detailed supplementary figures for the analysis of the parameters and stability of our method in the revised paper.

To Assigned_Reviewer_43:

Novelty: Although we construct our method based on the Indian Buffet Process, we describe a new generative story of a query in our work and extend it further for relevance feedback. We think the proposed generative model can be used with other Bayesian data models as well. In addition, it can be modified for different user needs in most retrieval systems.

Experiments: We chose to evaluate our method on the same datasets used in very recent relevant work by Rastegari et al. (ICML, 2013) because these datasets have publicly available pre-computed features and also class labels for quantitative evaluation. We thank the reviewer for referring the SBU Captioned Photo Dataset. Such a larger dataset will be very helpful to evaluate the effectiveness of our method.